# Global Cities in Transition: New York and Madrid in the Films of Chus Gutiérrez

Sagrario Beceiro , Begoña Herrero * , Ana Mejón and Rubén Romero Santos

Department of Communication and Media Studies, Universidad Carlos III de Madrid, 126, 28903 Getafe, Madrid, Spain; mbeceiro@hum.uc3m.es (S.B.); amejon@hum.uc3m.es (A.M.); rrsantos@hum.uc3m.es (R.R.S.)
* Correspondence: bhbernal@hum.uc3m.es

**Abstract:** In her triple condition of emigrant, artist and woman, the work of Spanish filmmaker Chus Gutiérrez is a privileged and singular object of study. Through her filmography it is possible to approach the changes that have taken place in the cities on both sides of the Atlantic. Chus Gutiérrez resided in New York during the decade of the 1980s and returned to Madrid to witness the changes that this city was undergoing: from being the epicenter of a dictatorship to a democratic city. Her vision of both places is clearly reflected in two of her films: *Sublet* (1992) and *El Calentito* (2005). The protagonists profiled by Chus Gutiérrez in these films are young women who move through complex metropolitan spaces at a critical moment of their lives. Cities become another character, illustrating, pushing, or limiting their course; spaces in which the protagonists accept their differences and begin the search for their individual and collective identity.

**Keywords:** city; Spanish filmmaker; immigration; globalization; Chus Gutiérrez; Madrid; New York; urban cultural studies





## 1. Introduction

Cinema's genesis is inextricably intertwined with the spaces and cultures of the modern city, as an ample body of scholarship has examined (Singer 1995; Clarke 1997). In tracing cinema's urban roots, researchers have elucidated how the visual technologies and spectacles emerging in metropolitan centers at the turn of the twentieth century informed the developing language of film. While cinema has now evolved into a predominately digital and globalized art form, cities remain pivotal sites where its lens continues to capture and reflect our contemporary stage of globalization. As Mennel (2008) notes, cities represent the foremost physical arena where processes of globalization are made manifest and registered by cameras. Cinematic representations of urban spaces offer a unique window into the flows, collisions, and inequalities of globalization. Compilations such as Pizza (2022) provide case studies of how film can be used to examine urban change, whether it is brought about by architectural transformations or the influence of tourism.

The evolution of cities has proceeded apace with developments in cinematographic arts. The most transformative urban shift in recent decades began in the late 1970s, as the oil crisis and decline of Fordist policies necessitated a reimagining of the modern city. Manufacturing bases dwindled as urban areas transitioned towards service economies and underwent drastic deindustrialization, a process led by major Anglo-Saxon cities that consolidated financial and corporate capital.

The term "global city", coined by sociologist Saskia Sassen in her seminal study *The Global City: New York, London, Tokyo* (Sassen 1991), is particularly successful in describing this process. Sassen points out in her study that changes in the nature of cities also imply a polarization of the labor market, which in turn corresponds to changes in the composition of their inhabitants and their way of life, with an increase in ethnic and cultural diversity.

Due to the cohabitation of clearly differentiated groups, debates on the nature of cities soon flourish. Under these new heterogeneous urbanisms arose contentious debates over

public and private space, gentrification, touristification, and urban citizenship. Harvey's influential "rebel cities" thesis (Harvey 2013) claims that while globalized cities may enable elite power, their density can also foster greater organization and resistance amongst disadvantaged inhabitants. Echoing Sassen's observations about changing urban relationships, Harvey further contends that disadvantaged city residents have both the capacity and responsibility to resist elite power. For Harvey, such resistance is not merely utopian fantasy but grounded in the unique organizational potential enabled by urban density and diversity. Given that cities concentrate and bring together diverse constituencies, they provide fertile ground for oppressed groups to build solidarity through coordinated networks and articulate collective dissent. Harvey thereby perceives an immanent promise within the global city for alternative political mobilizations and transformation through strategic coordination among its least privileged inhabitants. The contemporary metropolis thus represents a pivotal locus where new formations of resistance can crystallize and contest dominant urban agendas.

Harvey's "rebel cities" thesis must be complemented by a feminist perspective that spotlights gendered power relations within the global city. The contemporary metropolitan landscape has arisen concurrently with the ascendance of third-wave feminism. Critics such as Kern (2020) synthesize sociological and feminist analysis to argue that urban planning has historically manifest a patriarchal logic, designing city spaces by and for men. A radical reimagining of urbanism is thus needed. Applying Judith Butler's theory of performativity Kern claims for a gendered, embodied experience of city life. The feminist right to "occupy the space" expresses a transition from personal to collective praxis, from asserting one's corporeal presence to reclaiming the polis for inclusive participation and belonging. Through coordinated acts of urban citizenship, women can challenge their symbolic marginalization to transform city spaces into sites of empowerment. This feminist topology recasts the metropolitan environment as a privileged arena for contesting gender exclusivity within the myriad struggles over the global city. It elucidates the necessity for contemporary urban social movements to integrate feminist aims within their visions of justice.

Finally, the nascent domain of Urban Cultural Studies provides theoretical foundations and multidisciplinary perspectives to enrich the study of cities as instantiated within diverse creative and cultural texts. Urban Cultural Studies represent an interdisciplinary perspective for examining how material and social changes become embedded within cultural imaginaries and artistic representations of the contemporary city. As physical urban landscapes evolve, so too do the symbolic spaces through which city life is mediated, understood, and contested. Cultural artifacts provide insight into how urban transformations become expressed, filtered, and negotiated at the level of collective mythologies, memories, and meanings.

Within this field, Fraser's *Toward an Urban Cultural Studies* (Fraser 2015) draws from Henri Lefebvre to analyze representations of the global city in artistic works like cinema. Equally, scholars such as Lindner (2009) and Pile (2005) elucidate how urban imaginaries take shape through manifold cultural representations across various media delving into cinematic views. Their research highlights how artistic depictions of cities function not as passive mirrors of urban realities, but active prisms refracting those realities into multidimensional visions that can reinforce or challenge dominant perceptions. Attending the interplay between material conditions, lived experiences, and symbolic constructions is thus vital for a holistic understanding of contemporary urbanism.

This study conducts a focused examination of two films by Spanish director Chus Gutiérrez—*Sublet* (1992) set in New York and *El Calentito* (2005) in Madrid—as emblematic case studies of audiovisual urban representations. These works were strategically selected given the paper's aim of elucidating the city as an additional protagonist that structures the filmic imaginary. Both films were shot during pivotal transitional moments in each city's history, contexts that inflect Gutiérrez's artistic visions. Attending to the interplay between the material conditions of each city and their aesthetic translation through Gutiérrez's

directorial lens will elucidate the role of creative mediation in refracting urban realities into particularized narrative worlds. The value of this comparative case study lies in its capacity to isolate and examine how broader processes of urban evolution and imagination manifest through situated artistic practice. This study aims to elucidate how identity mediates artistic renderings of the city during pivotal transitions. Analyzing Gutiérrez's unique perspective can reveal critical insights about the interconnected nature of urban change, cinematic representations, gender, and migration.

As an emigrant, artist and woman, Spanish filmmaker Chus Gutiérrez provides a uniquely insightful case study for examining cinematic representations of urban change in New York and Madrid. Her liminal position allows her films to capture multidimensional perspectives on how these cities have transitioned. Selecting Gutiérrez as an object of analysis aligns with emerging scholarship that considers how urban representations are inflected by the filmmaker's subject position. Thornham (2019) argues that studying the city in cinema requires an intersectional lens attentive to disparate imaginaries shaped by directors' differing social identities and experiences. Within Spanish cinema, female directors before Gutiérrez like Cecilia Bartolomé and Pilar Miró had already incorporated the urban landscape as an essential storytelling element. Yet Gutiérrez's specific position as an immigrant woman director affords her films a particular nuance in portraying urban realities. Her outsider status strengthens the value of examining Madrid and New York through her distinct gaze.

Filmmaker Chus Gutiérrez rose to prominence in Spanish cinema during the 1990s as part of a pioneering generation of female directors shaped by Spain's political Transition (1973–1982). Gutiérrez belongs to a cohort of women filmmakers such as Gracia Querejeta, Isabel Coixet and Icíar Bollaín, who launched their careers during this postdictatorship cultural opening[1]. Coming of age amid seismic national changes, these directors produced new cinematic perspectives that reflected shifting social realities. Many obtained formative experiences abroad that further influenced their outlooks.

Though born in Granada in 1962, Chus Gutiérrez's family moved to Madrid when she was a child. She subsequently lived in London to learn English before immigrating to New York City at 21 to pursue filmmaking studies. Returning to Madrid in 1987, her early career reflects transnational encounters that shaped her multiperspectival outlook. As Barbara Zecchi notes, Gutiérrez's itinerant formative experiences imbued her work with a cross-cultural sensitivity attentive to the nuances of "otherness" and foreignness. As Zecchi points out:

> It should come as no surprise that this sensitivity towards the filmmaker's otherness derives from the fact of having constantly lived the experience of being "other": a Grenadian in Madrid, a Spaniard in England, a European in the United States; and a woman in the film industry. (Zecchi 2014, p. 165)

This analysis delves into an indepth comparative study of two key films within Gutiérrez's broader oeuvre: *Sublet* (1992) and *El Calentito* (2005) (Figure 1). Her first feature, *Sublet*, the only one he ever shot in New York, demonstrates Gutiérrez's initial efforts to portray the cityscape. The film is notable for its extensive location shooting depicting marginalized communities of New York during the 1980s. By contrast, the later *El Calentito* explores Gutiérrez's Madrid during the post-Franco cultural flowering of "La Movida". Her evolution as an artist is evidenced by *El Calentito*'s creative maturity and nuanced exploration of the social and musical counterculture emerging in Madrid in that era. These films were strategically selected based on their richness as case studies and embodiment of core themes in Gutiérrez's rendering of urban spaces undergoing transition.

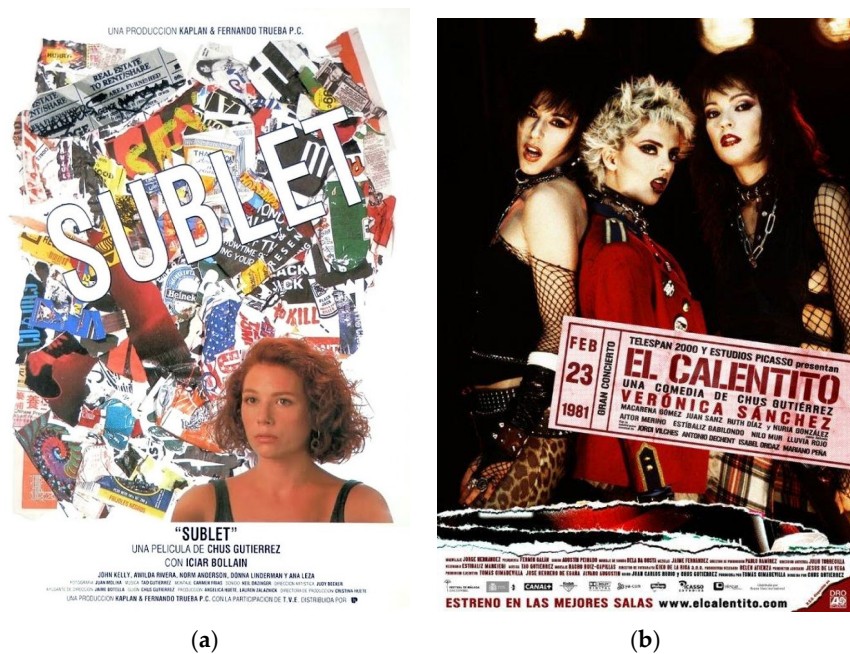

(a)          (b)

**Figure 1.** Release posters Chus Gutiérrez films: (**a**) *Sublet* (1992) Source: Fernando Trueba PC (https://fernandotrueba.com, accessed on 22 November 2023); and (**b**) *El Calentito* (2005). Source: Vertice Cine (https://vertice360.com, accessed on 22 November 2023).

## 2. Results

### 2.1. The Representation of the Gentrification of 1980s New York in Sublet (1992)

Chus Gutiérrez's debut feature *Sublet* offers a semiautobiographical portrait of New York City in the 1980s, drawn from her own experiences as a Spanish immigrant. Produced by renowned director Fernando Trueba, the English-language film follows Laura (Icíar Bollaín[2]), a young Spanish lawyer undergoing an existential crisis who impulsively travels to New York and becomes enamored with the city's frenetic energy. Captivated by its boundless promise, she leaves her job in Madrid to restart her life in New York. Yet her romantic notions about the city, similar to other films (Figure 2), are challenged as she struggles to find an affordable apartment amidst cutthroat real estate exploitation and neighborhood volatility.

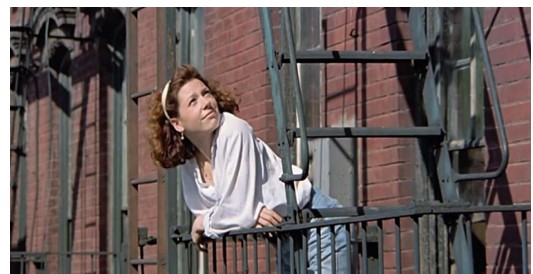

**Figure 2.** Chus Gutiérrez resorts to an iconic representation of the illusion of Laura (Icíar Bollaín), who dreams of being able to change her life in New York, just as Holly (Audrey Hepburn) does in *Breakfast at Tiffany's*. Source: Frame from *Sublet*. Chus Gutiérrez, 1992. Fernando Trueba PC (accessed at Filmoteca Española on 7 September 2023).

*Sublet* provides a street-level glimpse of New York's worsening socioeconomic polarization, aligning with seminal urban analyses like Sassen's *The Global City* and Mollenkopf and Castells' *Dual City*, both first published in (Mollenkopf and Castells 1991). Gutiérrez's gritty mise-en-scène illustrates these essays. It authentically captures the city's extremes, from opulent Park Avenue parties to drug-ravaged tenements, reflecting her own multi-

faceted experiences as an immigrant. Her protagonist's journey mirrors the aspirations and disillusions of those pursuing the myth of New York as a bastion of opportunity. Gutiérrez aesthetically renders both the alluring "light" of this mythos and its dark underbelly. As an emerging immigrant artist, her critical perspective on gentrification and inequality rejects romanticized visions of the global city to convey immediacy and nuance.

The film is set in Hell's Kitchen, a notorious example of gentrification. Located next to Manhattan's financial center, Hell's Kitchen was impoverished by deindustrialization in the 1980s, and exacerbated by its status as one of New York's main crack cocaine markets. The film takes place just as gentrification begins in the neighborhood, which includes renaming it Clinton. During the 1980s, the space became a target for financial speculation, leading to numerous altercations between tenants and landlords. The Windermere building became an iconic symbol of the greed of the latter and the resistance of the former. Similarly, the opening of the Jacob K. Javits Center in 1986, a convention center designed to sanitize the neighborhood by attracting the service sector and television studios, also fueled gentrification. As a result of these changes, Hell's Kitchen transformed from a neighborhood of working-class Irish immigrants, longshoremen, and slaughterhouse workers to a place where landlords seek to attract young artists. This gentrification process, as Saskia Sassen has argued, has the dual purpose of "sanitizing" the neighborhood of undesirable neighbors and increasing rents, as has happened in other parts of the city.

Gutiérrez subverts romanticized visions of New York by juxtaposing tourist enclaves against the daily struggles of marginalized residents (Figure 3). Her protagonist Laura contends with the city's extremes, from opulent wealth to tenement squalor, as she awakens to its internal contradictions. Gutiérrez herself recalls:

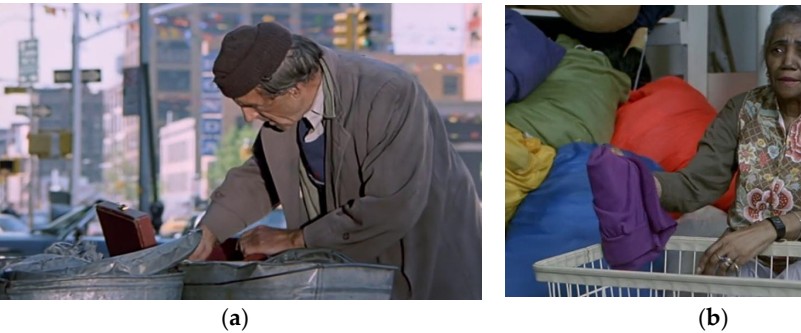

(a)  (b)

**Figure 3.** Gutiérrez mainly depicts marginal characters. Source: Frames from *Sublet*. Chus Gutiérrez, 1992. Fernando Trueba PC (accessed at Filmoteca Española on 7 September 2023); (**a**) A homeless man who lives on the street searches for valuables in the garbage; (**b**) Pakistani woman who runs the laundry where the protagonist does her laundry.

> It has incredible light! You wake up at six in the morning and the sky is blue. The city is light, it's not like London. [...] Suddenly one day you were at a millionaires' party on Park Avenue swimming in an indoor pool in minus ten degrees outside and another day you were dealing with a drug dealer on the Lower East Side. The social arc was brutal and everyone was curious. (Beceiro and Herrero 2019b, p. 156)

Yet she embraces these paradoxes as New York's essence, infusing the city itself with a mercurial personality that permeates each scene. Gutiérrez's recollections reveal her interest in overlooked spaces where the consequences of urban inequality become tangible. Her neorealist gaze bypasses postcard versions of New York to capture its more disquieting yet vital dimensions. As both filmmaker and former resident, Gutiérrez portrays neighborhoods like Hell's Kitchen not through a lens of decline, but as sites where new forms of community emerge in response to adversity. *Sublet*'s gritty urban textures ultimately expand the symbolic boundaries of what constitutes New York's identity as a global city:

I remember that in New York, at that time, there were many building owners who set fire to the buildings they owned in order to collect insurance... the building I lived in was very "heavy". It was a time when New York was very economically depressed, we are talking about '83 and well there were a lot of drugs, and people living on the street. (Beceiro and Herrero 2019a, p. 31)

Gutiérrez embeds her protagonist within the intricate social fabric of New York City, where a diverse cast of characters exhibit the pluralism and adversity beneath its cosmopolitan veneer. Laura's assimilation entails contending with an exploitative landlord, forging ties with immigrant communities, and navigating complex gender dynamics in her search for self-reinvention. Gutiérrez's nimble navigation of relationships across ethnic and class lines reveals the interconnected struggles of the urban poor. A melting pot of cultures represented by Gutiérrez, something evident in the film's soundtrack, which begins with the rhythm of *Io tengo n'appartamento*, by Renato Carosone. Her focus on female characters further explores issues of discrimination, domestic abuse, and solidarity amidst hardship. Though unable to find mutual understanding with her demanding male boss, she bonds with Carla, the female head of a moving company, through scenes metaphorically signifying women's collective strength. When Laura and Carla jointly carry a heavy futon upstairs, Gutiérrez creates microcosms signifying the collective strength women summon to survive urban precarity (see Figure 4a). Her protagonist's incremental empowerment parallels Gladys' ability to leave her abusive partner by accessing the support network around her (see Figure 4b). Gutiérrez's localized portraits cohere into a broader meditation on the quest for identity, community, and purpose within the anonymity of the global metropolis. Their capacity for self-actualization suggests that identity crises, though sparked by external factors like sexism, can be overcome through women reclaiming agency and purpose on their own terms.

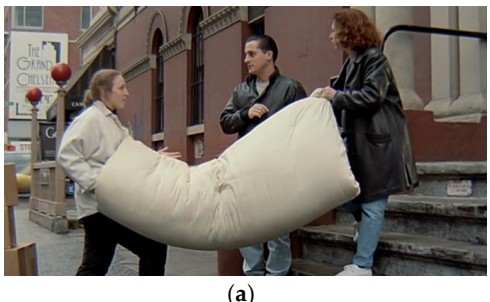 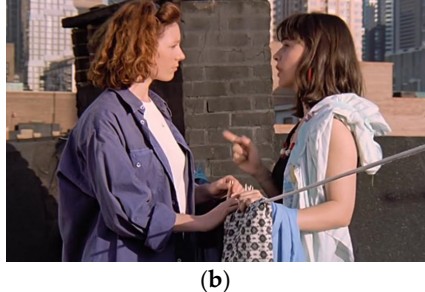

(**a**)                  (**b**)

**Figure 4.** The women form bonds and sisterhood relationships to survive in New York. Source: Frames from *Sublet*. Chus Gutiérrez, 1992. Fernando Trueba PC (accessed at Filmoteca Española on 7 September 2023); (**a**) Laura works for Carla at a moving company and they themselves carry heavy furniture up to high-rise apartments in walk-up buildings; (**b**) Laura supports Gladys in the face of her partner's domestic violence; in this frame, they converse on an iconic New York rooftop.

Beyond its sociological portrait, *Sublet* conveys Gutiérrez's intimate relationship with New York as a site of both disillusion and belonging. She affectionately describes it as a "shabby" yet captivating space that fosters devotion despite its pitfalls (Beceiro and Herrero 2019a, p. 34). Gutiérrez's lens captures the city's dual nature, at once isolating and communal, harsh, and alluring[3]. Her protagonist Laura experiences this duality, struggling with loneliness yet becoming profoundly anchored to the city. Gutiérrez's own attachments emerge through touches of the quintessential New York seen on postcards and movies, like aerial shots of the Manhattan skyline and a ferry ride towards the Statue of Liberty. These nods to its mythologized geography reveal her insider knowledge of the city's symbols (Figure 5). Ultimately, Gutiérrez's critical but affectionate gaze peels back the sheen of New York's globalized image to uncover the intimate human textures binding its disparate inhabitants. Her focus on overlooked communities resists romanticization while reasserting the dignity and hope sustaining the city's marginalized residents.

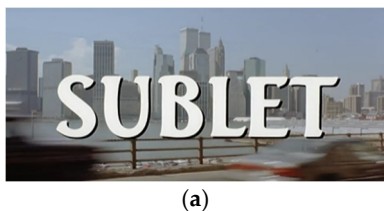
(**a**)
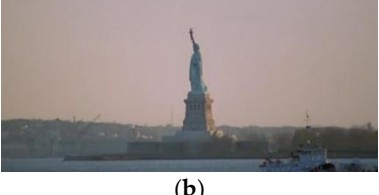
(**b**)
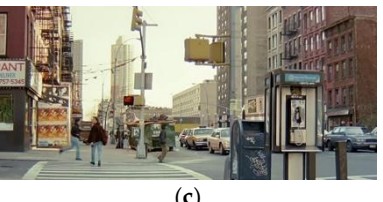
(**c**)

**Figure 5.** Iconic images of New York mingle with the stories of misery of the film's protagonists. Source: Frames from *Sublet*. Chus Gutiérrez, 1992. Fernando Trueba PC (accessed at Filmoteca Española on 7 September 2023); (**a**) Timelapse sequence of the New York skyline in the opening credits; (**b**) Statue of Liberty seen from a ferry; (**c**) New York Avenue.

In formal terms, *Sublet* exhibits an aesthetically daring visual sensibility for a debutant. It portrays New York City as a dynamic character shaping its inhabitants. Gutiérrez's creative use of time-lapse cinematography in the opening credits elegantly condenses the city's perpetual motion into a vivid temporality that her protagonist must navigate.

In *Sublet*, Gutiérrez reframes symbolic constructions of New York by light and shadow. Escaping clichés of corporate businessmen and wide-eyed tourists, she uncovers humanity in overlooked spaces—the liquor store, the dingy apartment, the street corner hangout. Gutiérrez portrays artists not as cosmopolitan elites, but urban vagabonds enduring hardship and exploitation alongside immigrant communities. The disillusioned painter Uma, who sublets her dilapidated apartment to Laura before fleeing the city to Greece because she "can't stand the city anymore", represents this outsider experience. Despite material decay, Uma's creative production vitalizes her apartment, transforming it into a shelter for Laura's own personal transformation. Gutiérrez's focus on creativity arising from society's peripheries remaps New York as a place defined not by its skyscrapers or celebrity culture, but by the resilient expressiveness of its most vulnerable denizens. Her vision excavates the shared humanity beneath surface differences, reconceptualizing the global city as a space for redemption through pluralistic exchange.

Gutiérrez also acknowledges the obstacles the global city poses to creative flourishing. Laura's artist neighbors Eugene, a homosexual aspiring singer, and Carlos, also a Spanish expat, embody unfulfilled aspirations. New York's cutthroat competitiveness stifles Eugene's dreams of becoming an opera singer while Carlos abandons his scientific career for an unstable artistic pursuit in a dilapidated studio. Carlos exemplifies a not so bright future for Laura if she decides to stay in the city for good. See Figure 6.

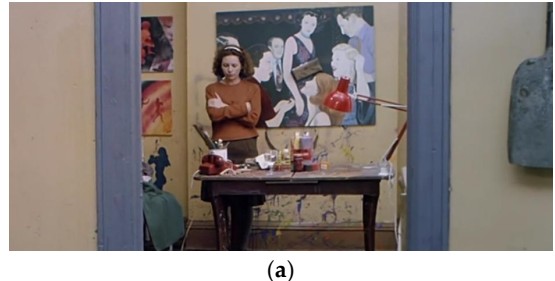
(**a**)
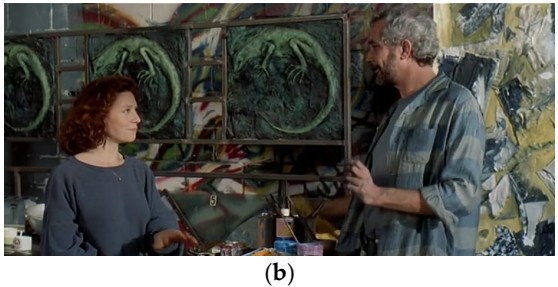
(**b**)

**Figure 6.** The artists have a relevant presence during the film, although they do not get the life of opportunities they dreamed of in New York. Source: Frames from *Sublet*. Chus Gutiérrez, 1992. Fernando Trueba PC (accessed at Filmoteca Española on 7 September 2023); (**a**) Laura feels safe in her apartment surrounded by Uma's creations; (**b**) Laura in the workshop of Carlos, sculptor, and painter.

Light and color play a fundamental role in delineating the personalities of the characters, particularly Laura's emotions, as her relationship with the city evolves. Despite the challenges she faces, Laura manages to fall in love with Alex, a young man from San

Francisco who spends a few days in New York. Alex represents Laura's escape from the pressures of precarious work and the intimidation of her landlord. Alex's return to San Francisco marks a farewell scene in which the passion between the characters is unleashed in a play of colors that will define the remaining aesthetics of the film. As Alex departs, the sadness of his departure darkens the image and bathes everything in blue, including Laura's face and the city itself. This scene is significant because it highlights the complex relationship between Laura and the city. The city is both a source of opportunity and a source of oppression for her. Alex's departure represents the loss of hope and possibility, and the city's blue hue reflects Laura's sadness and disappointment. See Figure 7.

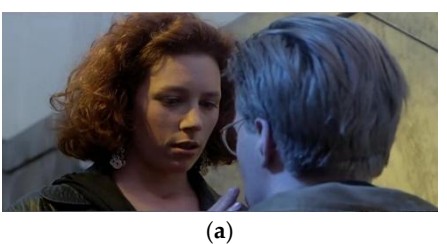
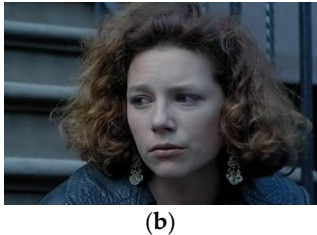
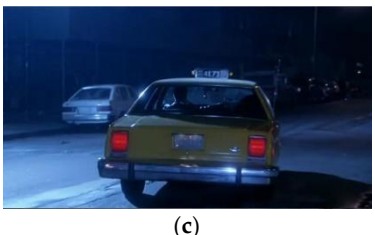

(**a**)         (**b**)         (**c**)

**Figure 7.** Gutiérrez uses lighting and color to identify Laura's emotions. Source: Frames from *Sublet*. Chus Gutiérrez, 1992. Fernando Trueba PC (accessed at Filmoteca Española on 7 September 2023); (**a**) Laura and Alex kiss in a play of red and blue lights before they say goodbye; (**b**) Laura is very sad after her lover's departure and depressed by the pressure exerted by her landlord. (**c**) New York becomes a blue city before the sadness of the protagonist.

Gutiérrez's urgent formal experimentation grows as Laura collapses under the weight of the city. New York is not ready for people like her, the city expels her. Gutiérrez employs the vagrant character to articulate the final meaning of this city for Gutiérrez: "With those eyes, what are you doing in this sewer? I am a rat. I live in sewers, below ground, and the rest of the world moves above. [...] They work, eat, sleep, and I get drunk", he tells Laura.

The vagrant's words are a stark reminder of the harsh realities of life in New York City. They also reflect Gutiérrez's own sense of alienation and despair. He sees Laura as a kindred spirit, someone who is also trapped in a city that does not care about them.

The vagrant's words also foreshadow Laura's eventual fate. She is unable to survive in the city, and she is ultimately forced to leave. Gutiérrez's proposal is a desperate attempt to save her, but it is ultimately unsuccessful.

Gutiérrez's proposal is a powerful moment in the film. It highlights the dark side of New York City and the ways in which it can crush the dreams of those who come to it seeking a better life. It also speaks to the deep sense of alienation and despair that can be experienced by those who live in the city.

The film's final resolution is especially relevant. Sadness leads Laura to a state of limbo, where she must live between the helplessness of not overcoming adversity and the unwillingness to give up the beautiful city she knew during her vacation days. In an allegorical sequence, she is shown particularly disheveled, eating yogurt with her hands while wearing a T-shirt with the iconic slogan "I love New York" (see Figure 8a). She then sees herself reflected in the mirror that was always broken in the bathroom (see Figure 8b). This moment represents Laura's rock bottom, and it becomes clear that she must leave the apartment.

The mafioso landlord ends up vandalizing Laura's apartment because he does not accept that it is sublet by his tenant (see Figure 9a). When Uma returns from her vacation and discovers the state of her home, she throws Laura out of the apartment, leaving her personal belongings in the street.

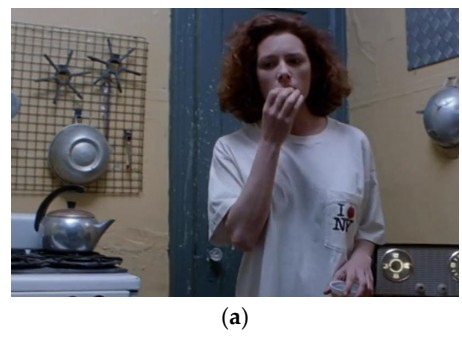
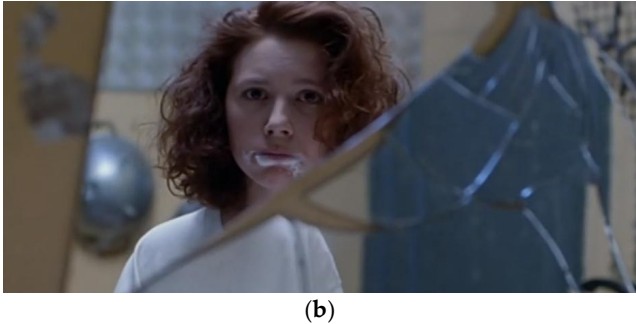

(**a**)                                                 (**b**)

**Figure 8.** Source: Frames from *Sublet*. Chus Gutiérrez, 1992. Fernando Trueba PC (accessed at Filmoteca Española on 7 September 2023); (**a**) Laura eats yogurt with her hands while wearing a promotional T-shirt of the city; (**b**) Laura's reflection in the broken mirror.

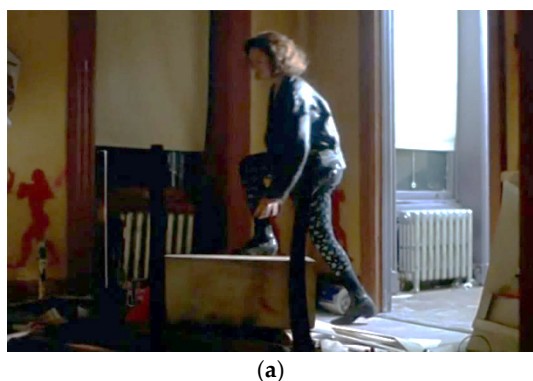
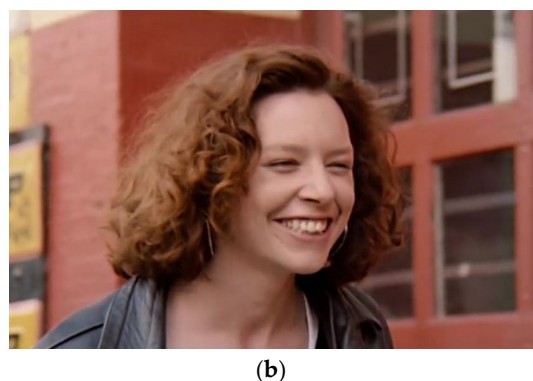

(**a**)                                                 (**b**)

**Figure 9.** Laura is expelled from her flat, but not out of the city. In an open ending, Gutiérrez confuses the viewer showing a relieved and smiling Laura. Source: Frames from *Sublet*. Chus Gutiérrez, 1992. Fernando Trueba PC (accessed at Filmoteca Española on 7 September 2023); (**a**) Laura discovers that the apartment has been vandalised; (**b**) Despite the adversity, Laura smiles at the end of *Sublet*.

Surprisingly, Gutiérrez plays with an open ending in the last sequence of the film. As he walks down one of the many indeterminate avenues of the city, the blue disappears and the red of the facades invades the frame. The city is once again vibrant, warm, just like Laura. A close-up of Laura's face shows the great disappointment she feels, while at the same time she smiles relieved, we do not know if it is because she is starting her return to Spain, or because she intends to continue living the adventure that awaits her in the unpredictable New York (see Figure 9b).

### 2.2. Freedom and Democracy in the Madrid of the Eighties in El Calentito (2005)

The second film under analysis, *El Calentito*, was shot in Madrid in 2005. It is a nostalgic and humorous look back at Spain in 1981, during the effervescence of the "Movida Madrileña" cultural movement. Unlike *Sublet*, which was shot shortly after the director's own experience of living in New York, *El Calentito* presents a different reflective process, as it was filmed 15 years after the events it depicts.

The film is set during the Spanish Transition (1975–1982), a period of profound political, social, and cultural change following the death of the dictator Francisco Franco. The so-called Transition was a complex and multifaceted process. It is one of the most interesting stages in Spain's recent history. It begins with the death of the dictator Franco in 1975 and the establishment of Democracy, although sociological change had already begun to occur since the previous decade.[4]

The arrival of socialist Enrique Tierno Galván[5] to the Madrid City Council in 1979 ushered in a period of modernization and transformation for the city, following the first municipal elections in Democracy. During these years, a number of urban planning

initiatives were proposed that would change the city's physiognomy. It was also a time of cultural ferment, known as the "La Movida Madrileña". As was happening simultaneously in other cities, Madrid was undergoing a metamorphosis: "It was a physical transformation, but also a conceptual one" (Romero-Santos and Mejón 2021, p. 31).

Mariví Ibarrola, a photographer whose work was recently featured in the exhibition *Yo disparé en los 80* ("I Shot in the 80s")[6], captured many of the protagonists of the Movida Madrileña on camera (Figure 10), including pop stars like Enrique Sierra[7] and Víctor Coyote[8]. In a recent interview, Ibarrola reflected on her work, saying, "When I was taking those photos, nobody knew that La Movida was going to be so important".[9] However, La Movida would become one of the most influential artistic and cultural movements in recent Spanish history.

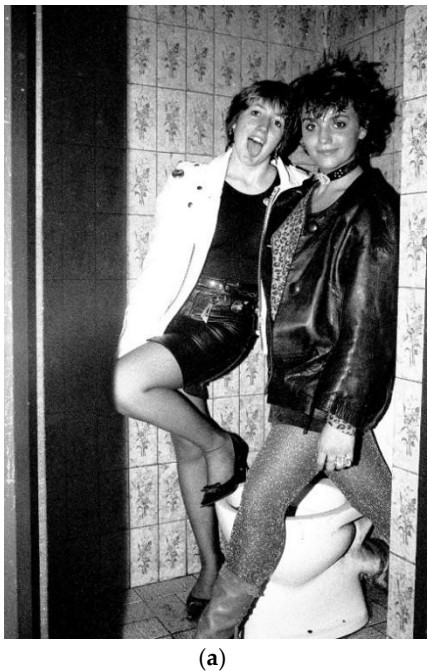 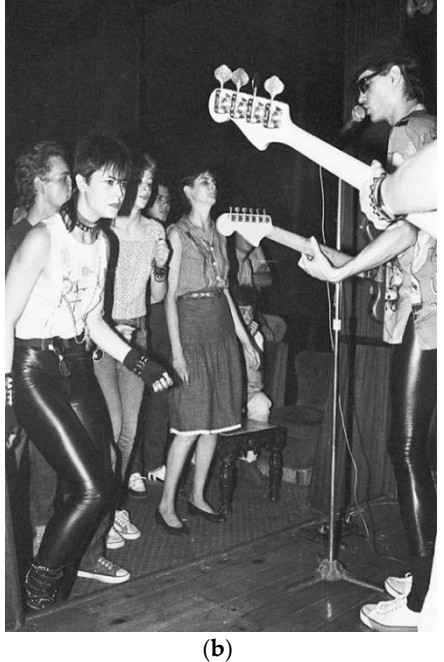

(**a**)                    (**b**)

**Figure 10.** Images taken by photographer Mariví Ibarrola in the city of Madrid, during the 1980s. Source: Mariví Ibarrola personal collection. (**a**) Image of two girls in punk attire in the bathrooms of the Sala Rockola, an emblematic concert hall of La Movida that was an essential meeting place for artists and musicians; (**b**) Audience attending a concert by Radio Futura, one of the most important musical groups of the time, at the Sala Rockola.

The 1980s in Spain were inevitably marked by the proximity of the Franco dictatorship (1939–1975). While Democracy came to Spain as an "unprecedented peaceful mutation", it also highlighted the urgent need for economic modernization and to address a number of political imbalances[10].

The cultural explosion of La Movida, which demanded freedom of expression, feminism, sexual freedom, and political freedom, coexisted with a Spain that still longed for dictatorship. This was evident in the conspiracies, military coup plotters, and citizens who loved the old regime. It was a historical moment in which, in addition to the exaggerated optimism generated by political change, discordant voices and more conservative social strata resounded. As a consequence of this circumstance, and at a time of deep political and economic crisis in Spain, on 23 February 1981 (known as 23-F), there was an attempted coup d'état that nearly ended the democratic Transition and the newly established parliamentary monarchy. The military coup was ultimately thwarted within hours, but it remains a stark reminder of the fragility of Democracy and the importance of vigilance against its enemies. While Democracy was taking its first euphoric steps, the tensions between statism and

renovation were about to explode on 23 February 1981 with the assault on the Congress of Deputies; a grotesque episode to be remembered could have turned into a serious crisis.[11]

The film *El Calentito* is set against the backdrop of the attempted coup of 23 February 1981, a watershed moment in Spanish history. The film explores the tensions between the old and the new, the traditional and the modern, in Spanish post-Transition Spain. On the one hand, the film represents the reactionary forces, those who wanted to go back to the regime of the military dictatorship. This is embodied by the character of Ernesto (Fernando Ransanz), a former police officer who is now a leader of the neofascist Alianza Nacional. On the other hand, the film represents the forces of progress, the next generation who are no longer even content with the achievements of Democracy and the Transition, but seek to achieve total freedom. This is embodied by the characters of punk music group "Las Siux": Leo (Macarena Gómez), Carmen (Ruth Díaz), and Sara (Veronica Sánchez).

Chus Gutiérrez returns to the years of the Transition in Spain, broadly understood, as they reconfigured the relationship between memory and cultural and political identity— both personal and collective (Herrero 2007, p. 79). Like her previous film, *Sublet*, *El Calentito* is a semi-autobiographical film with nostalgic overtones that brings to the present those turbulent years with irony and humor. Gutiérrez's close-up vision brings an unusual level of freshness to the story.

The film *El Calentito* tells the story of the punk music group "Las Siux", which performs in a venue in Madrid called "El Calentito", a real venue where Chus Gutiérrez herself worked. During this period, Spanish music was opening up to influences from the Anglo-Saxon world, as Triana Toribio (2017, p. 38) notes: Spanish punk/la Movida happened when influences from the UK and the US took root between 1977 and 1985 in a country that was enjoying a surge in freedom and creativity after the death of the dictator, particularly in the capital, Madrid.

The narrative of the film is suffused with the historical and sociopolitical milieu of the early 1981 period. The storyline revolves around the perspective of Sara, an introverted young woman hailing from a traditionalist family. Sara makes a conscious decision to relinquish her virginity, embarking on a fateful evening with her paramour to the establishment known as "El Calentito", wherein the musical ensemble "Las Siux" is scheduled to perform. Therein, she succumbs to inebriation, subsequently awakening at the residence of Carmen, a member of the musical group. Serendipitously, on the ensuing day, Sara is compelled to assume the role of a band member during an encounter with a music producer, a fabrication that inaugurates a profound personal transformation, ultimately prompting her to sever ties with her previous life.

The crux of the narrative unfolds on the eve of the band's pivotal performance, transpiring on the historically significant date of 23 February 1981. This date bears witness to a critical juncture in Spanish history, marked by the abortive military coup d'état that posed a substantial threat to the nascent Democracy that had recently been instituted in Spain.

The narrative of the film bears a marked autobiographical quality. In 1987, upon her return to Madrid from New York, Chus Gutiérrez embarked on a multifaceted professional journey that encompassed her involvement in a musical ensemble christened "Las Xoxonees"[12] along with her concurrent occupation as a waitress at the eponymous establishment that serves as the focal point of the film. Notably, this establishment was comanaged by Gutiérrez and her sister, the esteemed choreographer Blanca Li[13].

The musical entity, "Las Xoxonees" (see Figure 11) can be classified as a manifestation of the "post-punk feminist flamenco-rap" genre, as aptly characterized by Triana Toribio (2017). According to Gutiérrez's own recollections, the live performances of "Las Xoxonees" commenced in New York and seamlessly transgressed geographical boundaries upon their return to Spain. In Gutiérrez's own words, these performances embodied an amalgamation of entertainment, artistic expression, and levity, thereby aligning closely with the ethos of the "La Movida" movement: "Las Xoxonees" were fun and fresh, a mix of music, performance, provocation, and humor[14].

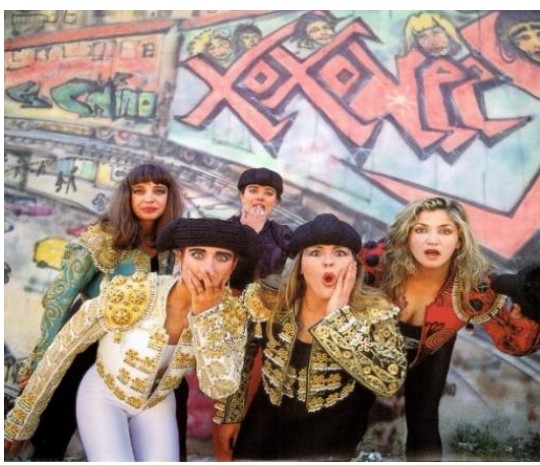

**Figure 11.** Promotional image of the group formed by Blanca Li (vocals), Chus Gutiérrez (vocals), Montse Martínez (vocals), Elena Robles (bass) and Silvia San Miguel (keyboard, bases, vocals). Source: Chus Gutiérrez & Revolution Lab. Available at: http://www.chusgutierrez.es/es/xoxonees (accessed on 18 July 2023).

Indeed, it is imperative to underscore that Gutiérrez did not merely function as a passive observer of the zeitgeist; she actively immersed herself in the musical milieu. This active engagement commenced in 1983, the inaugural year of the group's formation, when Gutiérrez became an integral participant in the vibrant musical landscape, initially in New York and subsequently in Madrid. About the birth of the group, Gutiérrez recalls with amusement:

> We were on 105th Street in New York and suddenly they said "let's start a group". We made two songs, out of nowhere. . . Because, also, if you came to see us in New York and we had a show, you would get into the show [. . .] We performed in a lot of places! Our first performance was so nice. . . we performed in a bar called The Blue Rose, which was on the corner of our street. A Greek woman wore it with a bun [. . .]. It was a seedy bar, no. . .the next thing. (Beceiro and Herrero 2019b, p. 157)

The director also talks about her return to Madrid in 1987, when she decided to return from New York and landed in the midst of the La Movida:

> I arrived in Madrid. I had nowhere to live. I had no way to earn a dime. I started working at the bar "El Calentito", which was a churrería on Jacometrezo Street, which belonged to my mother [. . .]. But of course it was open all night. He had a waiter, but he told us: "Why don't you guys work there?" It was a picture. . . all the hangers-on from Gran Vía came to us, the pimps, a picture. . . and the whole night! And then my sister Blanca thought that it was necessary to give it a radical turn and she kept the bar and set up "El Calentito". It was anthological. A wonderful place. Everyone from Madrid came. It had an ambigu downstairs and there was a little theater where Blanca performed with Paco Clavel[15]. Blanca's name was "SaraGosa"—a play on words with the name of the city Zaragoza and the phrase SarahEnjoy—and she ate a banana live. . . it was so fun! (Beceiro and Herrero 2019a, p. 35)

From a visual standpoint, the film *El Calentito* deliberately eschews the portrayal of the quintessential external landscapes emblematic of the city of Madrid. The narrative strategy adopted in *El Calentito* is distinctly characterized by a pronounced predilection for the depiction of characters and their intimately confined settings. These carefully delineated spaces serve not only as conduits for the narrative progression but also as potent vessels for the conveyance of the prevailing historical milieu. Notably, the film is punctuated by a paucity of sequences that unfold in open-air urban environs, with select

instances incorporating archival footage. It is essential to underscore that this intentional approach stands in stark contrast to the cinematographic treatment observed in *Sublet*, a film predominantly sited within the natural environs of New York City.

The historical milieu essential for contextualizing the cinematic narrative and the broader historical epoch of Madrid is subtly interwoven into the tapestry of *El Calentito*. This contextualization is predominantly manifest in the periphery of the characters, discernible within the intricacies of their attire, reverberating through their dialogic exchanges, and notably underscored in the meticulous ornamentation adorning their domestic habitats. It is within these contextual elements that we encounter the requisite historical substratum, serving as a narrative bridge that effectively binds the visual imagery with the collective imaginings of the era. This symbiotic relationship between visual representations and the construction of cultural imaginaries conspicuously elucidates the pivotal role that the metropolis of Madrid assumes as an additional, albeit intangible, protagonist within the cinematic tableau.

The film boasts a commendable feat of art direction under the stewardship of Julio Torrecilla. The domain of artistic direction within the realm of cinematic production is inherently entwined with the visual arts, encompassing disciplines such as painting, sculpture, and architecture[16]. Foremost among the notable achievements in this cinematic endeavor is the meticulously crafted backdrop constituting the domicile of the protagonist. This spatial construct effectively encapsulates the sociocultural milieu emblematic of a bourgeois and traditionally conservative Spain, still ensconced within the vestiges of the Francoist era (Figure 12).

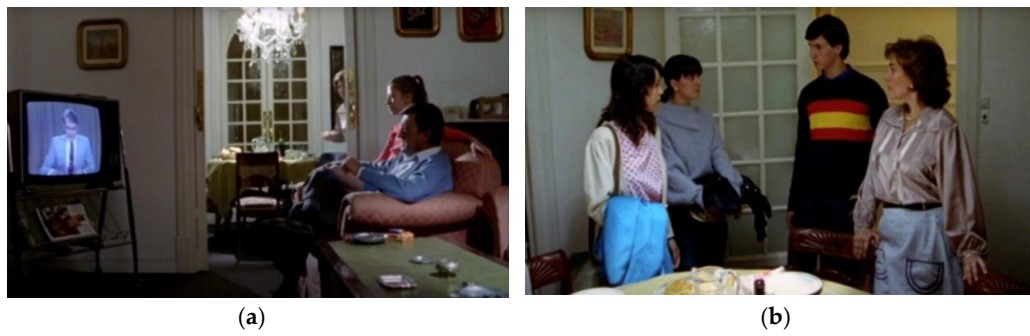

(**a**)  (**b**)

**Figure 12.** Two images of the protagonist's house representing an upper-middle class family home. Source: Frames from *El Calentito*. Chus Gutiérrez, 2005. Vertice Cine (accessed at Filmoteca Española on 4 September 2023); (**a**) detail of the living room with the television in the foreground and the sumptuous lamp in the background; (**b**) detail of the brother's clothing with the Spanish flag on his T-shirt.

Furthermore, the depiction of the music record company's environs is imbued with a sumptuous ostentatious and kitsch aesthetic, emblematic of the prosperity enjoyed by the prominent musical artists of the era, among whom Julio Iglesias holds a notable position. Concurrently, it serves as a visual testament to the ascendancy of La Movida groups that achieved remarkable acclaim during the cultural effervescence of the 1980s (Figure 13).

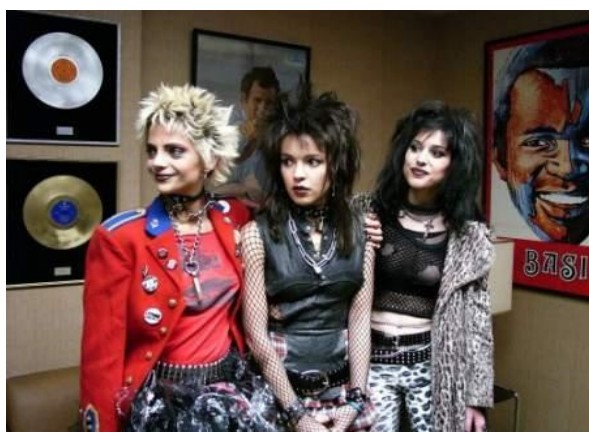

**Figure 13.** Image of the interior of the record company showing a space decorated with gold and platinum records and in the background, posters of famous artists such as Julio Iglesias and Basilio. Source: Frames from *El Calentito*. Chus Gutiérrez, 2005. Vertice Cine (accessed at Filmoteca Española on 4 September 2023).

Lastly, in a contrasting juxtaposition, we encounter two distinct spaces. On one hand, there is the abode inhabited by the members of the punk musical ensemble, which artfully captures the iconographic essence of La Movida. On the other hand, we find the interior milieu of the "El Calentito" bar (Figure 14a,b), the hallowed precinct where the pivotal musical performances transpire, thereby encapsulating the defiant and transgressive ethos that animated a burgeoning generation.

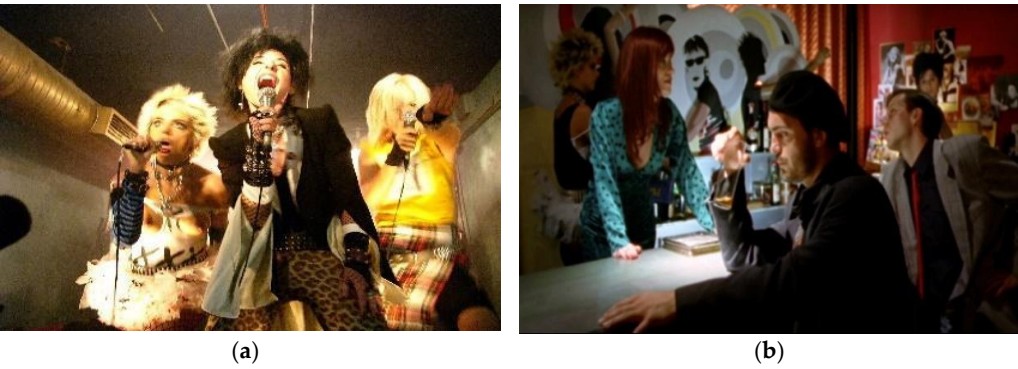

(**a**)                    (**b**)

**Figure 14.** Two images of the interior of "El Calentito". Source: Frames from *El Calentito*. Chus Gutiérrez, 2005. Vertice Cine (accessed at Filmoteca Española on 4 September 2023); (**a**) Detail of the stage during one of the band's performances; (**b**) The bar of the venue showing, on the red wall, a *collage* with photos of musical groups and concert tickets.

Chus Gutiérrez opts for a meticulous approach that centers on the delineation of interior spaces, wherein the narrative unfolds. This deliberate choice is poised to encapsulate the historical epoch through the prism of minute yet potent details, thereby engaging the viewer in a manner evocative of metaphorical discourse (Rodríguez-Cunill 2018, p. 553). These subtle details encompass, among others: the conspicuous presence of the Spanish national flag adorning both the room's interior and the attire of a character, (Figure 12b); the sartorial regalia of the musical ensemble's members, emblematic of punk subculture; references to ongoing news broadcasts addressing the activities of the terrorist organization ETA[17]; the presence of a Kung-fu poster within Carmen's living quarters (Figure 15a), belonging to one of the group members; and the visual inclusion of the newspaper cover heralding Spain's NATO accession within the narrative. Collectively, these elements converge to construct a nuanced metaphorical tapestry that elucidates the underlying essence of the historical milieu.

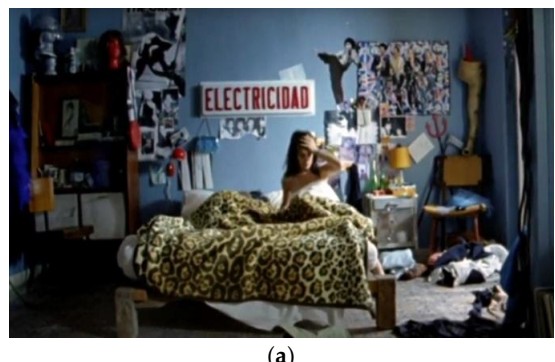 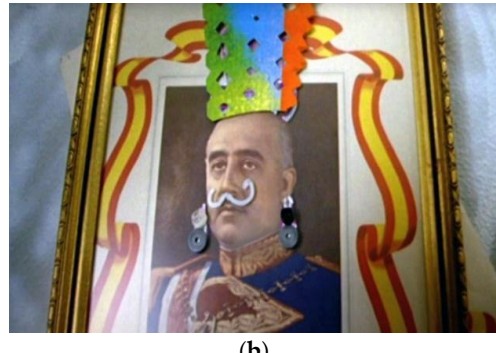

(**a**)                                                (**b**)

**Figure 15.** Space where the members of the punk group live. Source: Frames from *El Calentito*. Chus Gutiérrez, 2005. Vertice Cine (accessed at Filmoteca Española on 4 September 2023); (**a**) general shot of the group leader's room with cut-outs of the characters of the time pasted on the wall; (**b**) detail of a painting located in the hallway of the group's house, in which a caricatured portrait of Franco appears.

Conversely, the film employs a sparing utilization of exterior shots of the urban landscape of Madrid, primarily as transitional elements integral to the plot's narrative trajectory. These exterior shots encompass the descendent staircase and the opulent entrance of the residence in which the protagonist resides, notably featured when her paramour comes to visit. Notably, on the initial evening of her quest, the film strategically features an image of the entrance to the "El Calentito" venue (Figure 16a), depicting a throng of patrons queuing in anticipation of the forthcoming concert. Additionally, during the pivotal moment of the military coup d'état on February 23rd, the film incorporates a comprehensive panoramic view of the deserted city—a composition derived from archival footage—specifically capturing the urban panorama in the late afternoon of 23 February 1981, coinciding with the intrusion of Lieutenant Colonel Antonio Tejero into the Spanish Congress of Deputies (Figure 16b). These exterior shots, strategically interspersed within the film, serve as pivotal cinematic devices that punctuate and contextualize the narrative's progression.

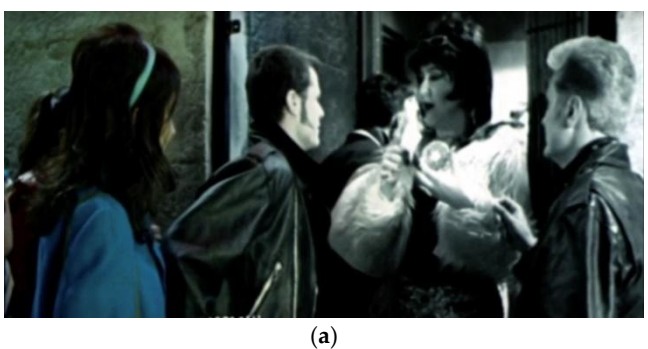 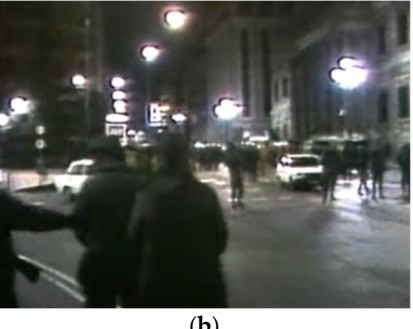

(**a**)                                                (**b**)

**Figure 16.** Two examples of exteriors shown in the film. Source: Frames from *El Calentito*. Chus Gutiérrez, 2005. Vertice Cine (accessed at Filmoteca Española on 4 September 2023); (**a**) The access door to "El Calentito"; (**b**) Archive image of the exterior of the Congress of Deputies located in Carrera de San Jerónimo on the night of 23-F.

Within the dramatic framework of the narrative, an acute temporal and spatial convergence is discernible, profoundly shaping the narrative trajectory. Notwithstanding its ensemble nature, the central narrative locus gravitates towards the emancipatory journey of a young woman hailing from a bourgeois family. In this context, the domicile of the protagonist assumes a conspicuous role, distinguished by an opulent and ornate decor, albeit tinged with claustrophobic undertones. The presence of a color television in the living area and an elegantly set dining table adorned with a pristine white tablecloth unequivocally bespeak the family's affluence.

Contrarily, the apartment inhabited by the members of the punk ensemble (Figure 15) and the "El Calentito" bar (Figure 14) epitomize representations of the lower strata of society, engendering a stark juxtaposition. This juxtaposition serves to accentuate the divergence between a luxurious and bourgeois spatial milieu, evocative of the erstwhile "old regime", and an urban, contemporaneous, liberated, somewhat dilapidated, and lower-middle-class universe.

It is pertinent to underscore that Gutiérrez's directorial vision refrains from relegating this lower-middle-class realm to a desolate and melancholic present, typified by squalid and desensitized environments. Rather, she invests these settings with a wealth of positive symbolic connotations, steeped in irony. Notable examples of this are manifest in the caricatured portrayal of dictator Franco (Figure 15b) and other emblematic facets. These spaces, thus imbued with a measure of irony, emerge as fertile grounds for the unbridled self-expression of the characters.

It is worth acknowledging that for viewers in 2005, the year of the film's release, the recollection of La Movida era predominantly encompassed a perception of playfulness and merriment. However, it is pertinent to underscore several overarching narrative threads (subplots) within the film. These narrative elements, coexisting within a cinematic representation of a liberated, hedonistic, and contemporary Madrid, deliberately evoke the darker undercurrents of that historical epoch. One of the salient subplots centers on the theme of homophobia, epitomized by the portrayal of Antonia, a transgender character who owns the bar. *El Calentito* actively probes the terrain of sexual diversity, encompassing depictions of transsexuals, homosexuals, bisexuals, and those who identify as queer. This thematic exploration constitutes an additional facet of the broader framework of otherness, in this instance manifesting as sexual difference, a motif recurrently observed throughout the cinematic works of Chus Gutiérrez[18].

At this juncture, it is inevitable, when addressing the context of the 1980s and the cultural movement known as La Movida in Madrid, to allude to the esteemed film director Pedro Almodóvar. A noteworthy curiosity lies in Almodóvar's contribution of archival footage for a specific sequence within the film (Figure 17). Almodóvar's cinematic oeuvre has been subjected to exhaustive analysis in a plethora of scholarly works. His cinema, which is notably rooted in the rejection of the left-leaning critics of the Franco regime (Smith 1994), presents a portrayal of a melodramatic Madrid that rediscovers its urban spaces in films produced during this epoch. Examples of such films include *Pepi, Luci, Bom y otras chicas del montón* (1980), *¿Qué he hecho yo para merecer esto?* (1984), and *La ley del deseo* (1987). Castejón Leorza et al. (2013) aptly noted in this context that Almodóvar's works, set against the backdrop of Madrid during "La Movida", serve as a reflection of the vibrant social, political, and cultural transformations underway within Spanish society.

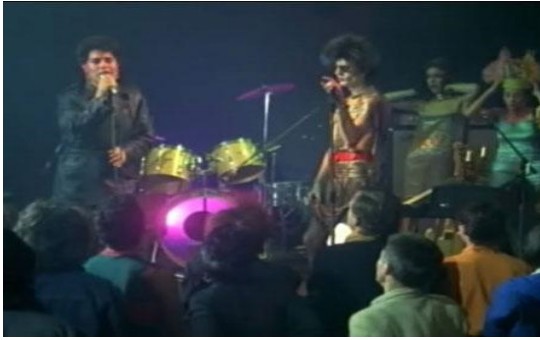

**Figure 17.** Mythical performance of Pedro Almodóvar and Fabio McNamara in a moment of the film. Source: Frames from *El Calentito*. Chus Gutiérrez, 2005. Vertice Cine (accessed at Filmoteca Española on 4 September 2023).

Having made this (obligatory) note, it should be noted that Gutiérrez in turn made his own portrait of the city of Madrid in the 1980s with *El Calentito*. The film somewhat

reproduces that atmosphere of the capital at that special moment. The film speaks about freedom as opposed to repression.

*El Calentito* does not merely depict the emergence of newfound freedom, as is often the case in works by Almodóvar, but instead, it presents a thematic underpinning where references to freedom serve as poignant contrasts to repressive forces poised to curtail it. Gutiérrez's directorial intent positions freedom as an ideal that becomes conspicuously salient in opposition to various repressive figures woven into the narrative fabric. These antagonists include, for instance, the mother of one of the protagonists, who thwarts her daughter's attendance at the concert, the morally corrupt police officer who employs coercion and extortion tactics, leveraging sexual favors, and the authoritarian neighbor who intrudes upon the premises during the night of the coup d'état, wielding a firearm and casting derogatory aspersions upon those present, labeling them as degenerates.

Throughout the narrative, there are scarce spaces that remain untainted by these pervasive repressive elements. The sole exception resides within the abode of the members of the musical group "Las Siux", which stands as a resolutely libertarian sanctuary. Even the establishment "El Calentito" faces imminent peril, ultimately prevailing against the odds owing to the unwavering resolve of the protagonists and their capacity to surmount adversity. It is not until the denouement, in the film's concluding sequence, that "El Calentito" unequivocally proclaims its status as an independent and libertarian enclave, dissociated from the shackles of the past.

This thematic trajectory is primarily mediated through the actions and agency of the characters themselves. A notable illustration of this is the pivotal scene in which the female protagonists defiantly sing, on the very night of the 23 February 1981 coup d'état, a song that had been previously censored. This act of protest, demanding freedom amidst the backdrop of political turmoil, concurrently catalyzes a personal journey characterized by the rejection of prescribed norms and the assertion of autonomous decision making as free individuals. Unquestionably, this narrative thread represents the most radical rupture within the overarching thematic tapestry.

In this multifaceted manner, the film serves as more than a mere tableau of a by-gone era; it emerges as a feminist manifesto and a resolute assertion of the conquest and reclamation of spaces.

## 3. Discussion

Several works partially address the figure of Chus Gutiérrez or some of her films[19]. These studies have predominantly centered on themes related to diversity and identity, particularly in the contexts of gender, ethnicity, and immigration. However, an examination of Gutiérrez's cinematic works as situated within the urban landscape, with a focus on the city as a pivotal and creative element, has not been a prominent focus in the existing scholarship. We contend that this perspective is of paramount significance in understanding the underlying dynamics of Gutiérrez's cinematic corpus.

In extensive interviews conducted by Beceiro and Herrero (2019a, 2019b), the director offered reflective insights into her formative years, particularly her sojourn in London upon reaching adulthood. During this period, she underwent a transformative phase, becoming acutely cognizant of the multifarious boundaries imposed upon her identity as a woman and a Spanish national, set against the backdrop of the intricate socio-political milieu prevailing during the transitional period between the 1970s and 1980s. This formative experience left an indelible imprint upon Gutiérrez's creative sensibilities and emerged as a salient leitmotif in her body of work.

Indeed, Gutiérrez's cinematic oeuvre consistently reflects her enduring preoccupation with the concept of boundaries in its expansive purview, encompassing not only their physical manifestations but also their functional implications. She demonstrates a sustained interest in the narratives of migratory movements, the complexities of cultural diversity, and the lives of individuals navigating diverse metropolitan environments. These thematic

currents collectively define the thematic tapestry that animates her cinematic universe. As Gutiérrez states:

> That was the first time I faced a border. They could not let me pass. . . and that changed me. I realized that we Spaniards were considered a poor country and that we were not welcome. He blew me away. I remember the feeling of fear, the world of the border. I also realized that the move from Granada to Madrid in my childhood had already been like an exodus, the first emigration. Then London, and then New York came too. That feeling of not belonging. . . (Beceiro and Herrero 2019b, p. 154)

Evident in both *Sublet* and *El Calentito*, the urban milieu emerges as a central and instrumental facet within the cinematic repertoire of Chus Gutiérrez. Across her cinematic works, Gutiérrez elucidates the premise that urban landscapes, by their very nature, serve as conduits and influential agents in the construction of the identities of their denizens. These identities encompass a sense of belonging to a communal entity and, conversely, the experiences of exclusion. This thematic exploration aligns with the perspective articulated by Mejón et al. (2019), asserting that identity inherently necessitates a contemplation of the intrinsic interplay between urban environments and facets of identity such as sexuality, class, and race.

Chus Gutiérrez's trajectory as an audiovisual artist distinctly bears the imprint of the cities in which she has resided, including Granada, Madrid, London, and New York. Significantly, these very urban centers serve as principal settings within her cinematic portfolio, with New York particularly featured as the locale for her debut film, and Madrid serving as an enduring nexus in her personal and professional journey as a filmmaker.

Simultaneously, the imperative of interweaving spatial representations with temporal dimensions emerges as a noteworthy facet, notably pronounced within the ambit of Chus Gutiérrez's cinematic oeuvre, and notably underscored in the films under examination, which aptly capture the zeitgeist of the 1980s. This confluence is instrumental in illuminating the integral role that cities assume in configuring distinct and temporally situated terrains for social coexistence. The narratives that unfurl within *Sublet* and *El Calentito* are imbued with a striking realism, encompassing the nuanced social advancements and challenges that are unique to each city and epoch delineated.

Moreover, this study delves into an exploration of Chus Gutiérrez's creative oeuvre to discern the diverse societal representations it engenders, within the overarching purview of Urban Cultural Studies. Specifically, it dedicates substantial attention to scrutinizing the evolutionary trajectory of Spanish society during the transitional years of democratic consolidation, as well as the countercultural expression embodied by the La Movida movement in Madrid. Through this analytical framework, Gutiérrez's films serve as dynamic windows into the multifaceted tapestry of sociocultural transformations and expressions that characterizes Spain during this epoch.

## 4. Materials and Methods

This study primarily relies on two primary sources. Firstly, the selection of films subjected to analysis adheres to the criteria elucidated within this exposition. While it is undeniable that the locales and urban settings featured prominently in Chus Gutiérrez's cinematic repertoire, *Sublet* and *El Calentito* have been chosen for examination due to their salient portrayal of the two cities where Gutiérrez spent her formative years, thus aligning with the sociocultural milieu of the 1980s. This decade holds paramount significance in the context of North American urban development and, concurrently, within the framework of the Spanish urban landscape, undergoing a profound transformation in the regime of civil liberties. The films were meticulously examined at the Filmoteca Española[20], an institution dedicated to safeguarding Spanish cinema and facilitating access for research purposes.

It is noteworthy that this selection (comprising *Sublet* and *El Calentito*) necessitated the omission of other works by the director, even though they similarly foreground urban life. Notable examples include *Alma gitana* (1996), which focuses on a specific Madrid

neighborhood and an ethnic community, and *Sacromonte, the wise men of the tribe* (2014), a documentary that retrieves the historical memory of a Granada suburb, also underpinned by ethnic themes, thus delving into a region that had long been relegated to the periphery of mainstream attention. These locales have been the subject of scrutiny in prior scholarly inquiries[21]. However, *Alma gitana* was excluded from consideration due to its collaborative nature, with input from multiple screenwriters, rendering it less emblematic of Chus Gutiérrez's distinct creative and personal milieu. Meanwhile, *Sacromonte, the wise men of the tribe* was omitted owing to its classification as a documentary, which diverges from the thematic orientation of the preceding films, and its comprehensive historical narrative, encompassing periods during which the author was not in residence in the city.

Furthermore, Gutiérrez's more recent productions, namely *Without You I Can't* (2021) and *From Little Red Riding Hood to Wolf* (2022), both set in Madrid, were excluded due to the inherent complexity associated with scrutinizing contemporary urban environments. Nevertheless, it is evident that these films, along with Gutiérrez's broader cinematic repertoire referencing alternative stages or epochs in the historical evolution of specific cities, remain consonant with the overarching aim of this study, which seeks to elucidate how urban centers serve as integral channels in the shaping of the identities of their inhabitants, fostering a sense of communal belonging. Consequently, these works may be considered for inclusion in subsequent, more expansive analyses.

Secondly, we conscientiously direct our attention to the comprehensive reference text, *En la frontera. Entrevista con Chus Gutiérrez* (Beceiro and Herrero 2019a), which is accessible online. Within this extensive interview, the director expounds at length upon her personal encounters and experiences within the cities of primary focus, namely Madrid and New York. Additionally, she provides intricate insights into the intricate aspects of the filmmaking process for the selected films under examination.

The process of analyzing the films adopted a structured methodology encompassing the following key components:

- Scene Selection: The initial step entailed the meticulous curation of scenes featuring exterior urban settings within the cities of Madrid and New York. Subsequently, efforts were directed towards the identification of specific locations, whenever feasible.
- Narrative Analysis: A comprehensive narrative analysis was conducted, concentrating on the dynamic evolution of both primary and secondary characters. Particular emphasis was placed on their interactions with and adaptation to the urban milieu. Notably, this analysis encompassed an exploration of immigration-related themes within the context of *Sublet* and an examination of the historical transitional period to democracy within the purview of *El Calentito*.
- Formal Analysis: An indepth formal analysis was undertaken, focusing on the proficient employment of cinematic techniques. This facet of the analysis sought to discern the cinematic craft's role as a medium for portraying urban locales and to ascertain its impact on the characters' development throughout the narrative trajectory.

This approach, which encompasses an extensive exploration not only of the formal aspects inherent to cinematic works but also their intricate interplay with the director's personal biography, holds the potential to be adopted in future scholarly inquiries. Such a methodology enables a profound examination of the unique perspectives of diverse audiovisual creators in the intricate construction of urban imaginaries within the realm of cinema.

## 5. Conclusions

The films under scrutiny encapsulate an approach that regards cities as entities imbued with a distinctive character of their own. Within this paradigm, cities emerge as dynamic constructs, constituting a specific temporal and spatial milieu that not only serves as a backdrop but also actively shapes and configures pivotal aspects of the lives and experiences of their inhabitants. Central to this framework is the nuanced exploration of the notions

of community, individual identity, and the interplay of factors such as sexuality, class, and race.

This thematic exploration finds poignant manifestation through the character of Laura in *Sublet*, whose transformative journey unfolds within the backdrop of New York City. Her sojourn in this urban center fundamentally reshapes her worldview, thus catalyzing a profound shift in her perception of life, a transformation irrevocably etching her existence. Concurrently, the characters in *El Calentito* derive their essence and disposition from their spatial and temporal context, notably underscored by the emblematic libertarian space previously delineated. Situated within the burgeoning cultural effervescence of La Movida in Madrid, these characters are intrinsically products of their environment and the era they inhabit.

A recurring thematic motif that underscores both films is the palpable and pressing desire exhibited by their characters to transcend their present circumstances, to transition into alternate spaces that symbolize a departure from the constraints of their past. This shared impetus for personal progress is a palpable driving force animating the trajectories of the characters in both films.

In *Sublet*, New York City is portrayed as a seemingly inhospitable urban landscape where immigrants like the protagonist grapple with the periphery of society. Meanwhile, *El Calentito* captures Madrid in a state of nascent emancipation, a city on the cusp of embracing freedom, albeit tinged with the inherent risk of regressing due to the presence of repressive elements intent on constraining the trajectory of that evolution and curtailing the burgeoning sense of freedom that it embodies.

In summary, it is evident that within the narrative frameworks of *Sublet* and *El Calentito*, the urban settings in which these stories transpire, encompassing neighborhoods and distinct locales, stand as inextricable elements intertwined with the developmental trajectories of the characters. The protagonists, delineated by the director Chus Gutiérrez, traverse intricate and multifaceted urban landscapes, which play a multifaceted role in shaping their choices and defining their modes of existence. This dynamic interplay is underscored by the notion that these urban environments, on occasion, facilitate the characters' journeys, at times exert constraints upon them, and at other junctures, propel them forward in their respective narratives.

In a profound articulation of thematic intent and from an overtly sanguine vantage point, Gutiérrez adeptly presents two divergent cities—New York and Madrid—spanning two disparate continents, each grappling with the complexities of their respective historical epochs. These urban spaces, while divergent in their geographical and sociocultural contexts, share a common thread: they serve as arenas wherein the characters ultimately discover the capacity to coexist harmoniously, embracing distinctions, fostering the liberty of self-expression, and embarking upon the profound journey of self-discovery.

**Author Contributions:** Conceptualization, S.B. and B.H.; Formal analysis, S.B., B.H., A.M. and R.R.S.; Funding acquisition, S.B., B.H., A.M. and R.R.S.; Investigation, S.B., B.H., A.M. and R.R.S.; Methodology, S.B., A.M. and R.R.S.; Project administration, S.B. and A.M.; Supervision, S.B.; Validation, B.H., A.M. and R.R.S.; Visualization, S.B., B.H., A.M. and R.R.S.; Writing—original draft, S.B., B.H., A.M. and R.R.S.; Writing—review & editing, S.B., B.H., A.M. and R.R.S. All authors have read and agreed to the published version of the manuscript.

**Funding:** This research was funded by Research Project I+D+i "Cine y televisión en España en la era digital (2008–2022): nuevos agentes y espacios de intercambio en el panorama audiovisual" (PID2022-140102NB-I00/AEI/10.13039/501100011033), Agencia Estatal de Investigación, Ministry of Science and Innovation/Ministerio de Ciencia e Innovación, Government of Spain/Gobierno de España. Also it had the support of the Project "Memoria oral de las mujeres en el cine español de la Democracia" (Ref. 35-7-ID22), Women's Institute/Instituto de las Mujeres, Ministry of Equality/Ministerio de Igualdad, Government of Spain/Gobierno de España.

**Data Availability Statement:** Data are contained within the article.

**Conflicts of Interest:** The authors declare no conflict of interest.

## Notes

1. The period of transition from Franco's dictatorship to democracy in Spain and the way the media were transformed in those years has been extensively studied by the Tecmerin research group of the University Carlos III of Madrid in works such as *El cine y la transición política en España* (1975–1982) (Palacio 2011) and *Cine y cultura popular en los noventa* (Palacio and Rodríguez 2020).

2. Icíar Bollaín: although she began her career in film as an actress, has subsequently developed a significant career as a film director in Spain.

3. This perception coincides with the one developed by Mehta (2017), who explores a portrait of cities through the experience of the immigrants who inhabit them.

4. Other perspectives on the social changes that took place during the transition to democracy in Spain can be expanded in the works of Palacio (2011), Juliá (2017) or Mainer and Juliá (2000).

5. "35 años sin Tierno Galván, el alcalde enrollado que pedía a la gente que se colocara". Available at https://www.elmundo.es/loc/famosos/2021/01/19/6005d219fc6c83cd188b45dc.html (accessed on 11 august 2023).

6. This exhibition has been part of PHotoESPAÑA 2023, one of the most important international photography meetings. In the XXVI edition in 2023, 303 photographers and visual artists from different countries will participate, addressing three curatorial axes: art, environment and gender.

7. Enrique Sierra Egea, Spanish musician and composer, was the founder of some of the most representative groups of "La Movida" such as Kaka de Luxe or Radio Futura.

8. Victor Coyote was a multifaceted artist, very active during the years of "La Movida": graphic designer, illustrator, painter, musician, comic book author, writer and documentary and video clip maker.

9. Available at: https://www.laopinioncoruna.es/coruna/2023/08/30/marivi-ibarrola-fotos-nadie-sabia-91485092.html, accessed on 25 August 2023.

10. Further information about this period can be found in Cotarelo (1994) and Pinilla García (2021).

11. The assault on the Congress of Deputies in Spain on 23 February 1981 has been extensively explored by Quaggio (2014) and Ruiz (2006).

12. The Xoxonees started in New York in 1983. They defined their style as Flamenco-Rap and their lyrics were full of humor, irony and criticism. In 1986, some members of the group began to return to Spain, and it was in 1987 when the group was recomposed again and began a more professional career. They performed regularly in "El Calentito", the venue run by Blanca Li, but they also performed in other venues and places in Spain. Available at: http://www.chusgutierrez.es/es/xoxonees (accessed on 12 July 2023).

13. Blanca Li, born in Granada in 1964, is a choreographer, dancer, actress and film director. She has choreographed and directed numerous ballets, operas and musicals, including "Les Indes Galantes" at the Paris Opera, "Shéhérazade" for the Paris Opera Ballet, "Very Gentle" and "Un Parque" by Luis de Pablo, at the Teatros del Canal or "Treemonisha" at the Théâtre du Châtelet in Paris.

14. See Gutiérrez website: http://www.chusgutierrez.es/es/xoxonees (accessed on 18 July 2023).

15. Paco Clavel was a popular pop artist and singer during "La Movida", promoter of the "cutre-lux" aesthetics, accompanied by a striking aesthetic. In 1980 he was part of the musical group Bob Destiny & Clavel y Jazmín.

16. Inmaculada Rodríguez-Cunill (2004) explores the connections between the arts and certain film professions such as art direction, set design, make-up and costume design.

17. ETA Militar was a Basque terrorist group with a pro-independence ideology that operated in Spain from the 1960s until its dissolution in October 2011.

18. The presence of sexuality in the work of Chus Gutiérrez has been extensively referenced by Barbara Zecchi (2014). In particular, the case of *El Calentito* and its rich representation of sexual diversities has been studied by Bartolomé (2023).

19. We refer to the works about Gutiérrez as filmmaker, developed by Camí-Vela (2001), Colaizzi (2007), Everly (2016), Zecchi (2014), Molano (2015); or those focused on some of her films such as Nair (1999), Parejo (2014), Corbalán (2015), López-Cabrales (2019), Bartolomé (2023).

20. Filmoteca Española is a public organization in charge of preserving Spain's cinematographic heritage. Its mission is to recover, research and preserve the cinematographic heritage and promote its knowledge. The film collection is the core of Filmoteca Española and its raison d'être. https://www.culturaydeporte.gob.es/cultura/areas/cine/mc/fe/portada.html, accessed on 15 September 2023.

21. The references on the representation of the city of Madrid and its neighborhoods are extensive, both from more historical perspectives such as that of Deltell (2006), as well as from approaches to specific works, such as that of López-Cabrales (2019) in the analysis of *Alma Gitana* or *Carmen y Lola*, of special interest for this work. Representations of Granada, the city in which *Sacromonte* takes place, have also received scholarly attention, with Puche Ruiz and Gámir (2023) as one of the most recent.

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
