# Peer review of "Global Cities in Transition: New York and Madrid in the Films of Chus Gutiérrez"

_arts, 1999_

Round 1
Reviewer 1 Report
Comments and Suggestions for Authors
I would like to propose a minor revision to the authors: it pertains to the expansion of the bibliography to include recent works related to the proposed article, such as, for instance, the book "La Ciudad en el Cine," by Antonio Pizza (ed.). Furthermore, there is a need to rectify the alphabetical order of one of the references (Nair, 1999), which is not accurately positioned. Beyond that, I find no issues or errors in the proposed article, and I believe that the analysis exhibits commendable quality.
Author Response
We appreciate all the considerations made by reviewers to improve our work. Here, we present the changes made throughout the document, as well as clarifications to some of the questions posed by the reviewers.
All the changes are marked in red in the text.
REVIEWER 1
- The reference “La Ciudad en el Cine” by Antonio Pizza (ed.) was included.
- The reference (Nair, 1999) was repositioned
Reviewer 2 Report
Comments and Suggestions for Authors
The article "Global Cities in Transition: New York and Madrid in the Films of Chus Gutiérrez" is a remarkable and thought-provoking piece of research that delves into the multifaceted relationship between urban transformation, film, and identity. Chus Gutiérrez's unique perspective as an emigrant, artist, and woman is skillfully explored, making this study a compelling read.
The introduction provides an insightful background on the evolution of cities in the context of globalization, offering a comprehensive view of the global city concept. It highlights the impact of urbanization on labor markets, inhabitants, and the complexities of urban life. The mention of David Harvey's "Rebel Cities" thesis and the need for feminist perspectives within urban studies adds depth to the discussion.
The inclusion of Urban Cultural Studies as a theoretical foundation is commendable, as it emphasizes the importance of understanding how cultural representations in various media intersect with urban realities. The authors’ examination of the two films, "Sublet" and "El Calentito," is a good choice, as they were shot during pivotal moments in the history of New York and Madrid, adding a historical context to the analysis.
The article effectively emphasizes Chus Gutiérrez's unique position in the world of cinema and her role as a pioneering female director. The exploration of her immigrant background and experiences in New York and Madrid, and how they inform her cinematic representations, is an engaging aspect of the study.
Furthermore, the article successfully captures the essence of Chus Gutiérrez's films as an embodiment of the cities themselves, illustrating how cities become integral characters in her narratives. This approach enriches our understanding of the interplay between urban change, cinematic art, and identity.
The article's depth and well-structured arguments make it a valuable contribution to the field of urban studies, cultural studies, and gender studies. This research is essential for those interested in understanding the dynamic relationship between urban spaces and artistic expression.
I highly recommend "Global Cities in Transition: New York and Madrid in the Films of Chus Gutiérrez" for publication with no changes, the article is an enlightening and comprehensive exploration of the intersection of urban change, cinema, and identity. We consider the manuscript to be well-researched, logically structured and offer a unique perspective that will inspire further research and discussions in the field.
Thanks.
Author Response
We appreciate all the considerations made by reviewers to improve our work.
Reviewer 3 Report
Comments and Suggestions for Authors
Although there is evidence throughout of extensive relevant research and textual knowledge, this article requires very extensive additional work to bring it up to publishable standard. I have listed the issues which need addressed below, nearly all of which relate to structural problems and a significantly underdeveloped argument:
· A clear introduction needs to be in place which provides a very precise overview argument which ties together the analysis of the two films. What questions is this article asking about Gutiérrez’ work, specifically about the different urban landscapes and period of cultural history being depicted? How can urban theory help us address these questions? Does Gutiérrez have a distinctive urban aesthetic which carries across the two films or is there something quite different being shaped in the language of each film? As it stands the introduction is too broad and fragmented. What does it mean that cinema has its origins in urban landscapes (without clear reference points)? Lumiere and Méliès, for example, are quite different in this respect. Followed by a jump to an unrelated but equally large claim about globalisation. In short, the article needs a coherent overview at the start which established the terms of the discussion of the two films in question and how this will be framed with reference to theories of urban development in a globalised world.
· Argument starts to become more focused here on p.2: ‘This study conducts a focused examination of two films by Spanish director Chus Gutiérrez—Sublet (1992) set in New York and El Calentito (2005) in Madrid— ‘. This might be a better place to start? But what does it mean to argue that the city is a protagonist? This surely requires analysis? What sorts of agency can it possess? Doesn’t this contradict the Butler related point above human\female agency within the city?
· There are too many short undeveloped paragraphs which are indicative of an argument which is reluctant to open up key details and push its thinking on key issues. Many assertions beyond the texts are unsupported by references outwards to academic work which supports the claims being made.
· Style throughout requires further editing for precision and economy. e.g ‘as aforementioned’ p.3
· Longer quotations – not indented.
· More detailed discussion of film language need to be included. For example: ‘Yet she embraces these paradoxes as New York’s essence, infusing the city itself with a mercurial personality that permeates each scene’. p. 5 How? More analysis of evidence from the film is needed – discussion of lighting, mise-en-scene etc. Testimony from the film maker is not evidence about the film itself.
· p.7 ‘This stylized establishing shot announces Gutiérrez’s interest in moving beyond surface im- pressions to uncover the deeper kinetic rhythms of urban life’. Needs to unpack what these rhythms are and what is revealed beyond the surface.
· Mentions a defining aesthetic of colour and light but only provides one example p.8
· Much stronger sense of an ongoing argument needs to be in view throughout. What is at stake in the comparison between these films? Why is it important to compare them? We need to keep getting answers to these questions.
· The discussion of El Calentito from p.10 suffers from the same levels of under development. There’s 3 pages of context before it gets to the film and doesn’t set up the points of distinction and connection to the previous one. It picks up a connection on p.13 but it doesn’t quite make sense because of the use of the term naturalistic. Surely interiors can also be naturalistic – does this rather refer simply to nature/natural?
· p.15 ‘These exterior shots, strategically interspersed within the film, serve as pivotal cinematic devices that punctuate and contextualize the narrative's progression’. Needs to explain/chart what the nature of this progression is in more detail.
· The reference to Almodóvar feels a bit grudging. It doesn’t fully situate his work in relation to Gutiérrez so is a bit redundant. The bigger point about Almodóvar only representing the forces of freedom, not repression, seems excessively broad.
· Discussion section is a form of literature review which should come at the start of the article.
· Likewise the materials and methods section – both these last two sections need to be edited down with a lighter touch (and removal of the section with bullet points).
Comments on the Quality of English Language
Good, though editing for precision required.
Author Response
We appreciate all the considerations made by reviewers to improve our work. Here, we present the changes made throughout the document, as well as clarifications to some of the questions posed by the reviewers.
All the changes are marked in red in the text.
REVIEWER 3
We have done our best to include as many remarks as possible. However, some observations reference the structure requested by the journal and cannot be done without modifying their pattern. As for those who can be done:
Regarding the introduction:
The first paragraphs of the introduction have been changed to clarify how global cities have developed since the 70s and why we chose these two case studies. How Chus Gutierrez ties with the topic is implied when talking about her triple condition as a woman, immigrant, and artist. Commentary and references were added about urban development and the birth of cinema.
The sentence started with “as aforementioned” has been reduced to: This analysis delves into an in-depth comparative study of two key films within Gutiérrez's broader oeuvre.
Regarding film analysis:
We think the selection of these two films is explained in the text, specifically in the last paragraph of the introduction and in the methods part. They are linked by the representation of two big cities in which Gutierrez lived in times of social or political transition.
The paragraphs on page 5 related to Hell’s Kitchen were unified.
Longer quotes were indented and marked in red.
The sentence example: ‘Yet she embraces these paradoxes as New York’s essence, infusing the city itself with a mercurial personality that permeates each scene’ on page 5 is a literary resource to introduce the analysis of the film. The reviewer asks “how” is it done. The answer is just in the next sentences and during the analysis: by mixing some of the most iconical (touristic) images of New York with images of the daily life of specific residents. We specifically talk about “urban inequality”, and the representation of “neighborhoods like Hell’s Kitchen not through a lens of decline, but as sites where new forms of community emerge in response to adversity”.
The reviewer requires a more extensive analysis of the film. At least 7 sequences have been analyzed during the texts and more specifically in the foot of figures 2 to 9 following the style of the journal. The extension of the paper does not allow us to expand it.
In terms of considering the filmmaker's testimony, even though it shouldn't be regarded as evidence of the films themselves, we aim to demonstrate the parallelism between Chus Gutierrez's perception of these cities and her experiences with some of the situations depicted in the movies. The core focus of the article is not to convey a formal analysis, but rather to comprehend why this Spanish woman engages with these two representations.
In agreement with the reviewer, we suggest removing the sentence “This stylized establishing shot announces Gutiérrez’s interest in moving beyond surface impressions to uncover the deeper kinetic rhythms of urban life” as is repetitive with the previous idea.
On page 13, the word “naturalistic” changes to “natural” as noticed by the reviewer.
Regarding the “discussion” and “materials and methods” sections
They have been placed according to the journal style.

Round 2
Reviewer 3 Report
Comments and Suggestions for Authors
Although I still have concerns about the quality of the argumentation here I think the manuscript has been improved sufficiently to publish.